# Focused Ion Beam Milling of Single-Crystal Sapphire with A-, C-, and M-Orientations

**DOI:** 10.3390/ma13122871

**Published:** 2020-06-26

**Authors:** Qiuling Wen, Xinyu Wei, Feng Jiang, Jing Lu, Xipeng Xu

**Affiliations:** 1Institute of Manufacturing Engineering, Huaqiao University, Xiamen 361021, China; weixinyu163@126.com (X.W.); jiangfeng@hqu.edu.cn (F.J.); lujing26@hqu.edu.cn (J.L.); xpxu@hqu.edu.cn (X.X.); 2Fujian Engineering Research Center of Intelligent Manufacturing for Brittle Materials, Huaqiao University, Xiamen 361021, China

**Keywords:** sapphire, focused ion beam, crystal orientation, etching, material removal rate, surface roughness

## Abstract

Sapphire substrates with different crystal orientations are widely used in optoelectronic applications. In this work, focused ion beam (FIB) milling of single-crystal sapphire with A-, C-, and M-orientations was performed. The material removal rate (MRR) and surface roughness (Sa) of sapphire with the three crystal orientations after FIB etching were derived. The experimental results show that: The MRR of A-plane sapphire is slightly higher than that of C-plane and M-plane sapphires; the Sa of A-plane sapphire after FIB treatment is the smallest among the three different crystal orientations. These results imply that A-plane sapphire allows easier material removal during FIB milling compared with C-plane and M-plane sapphires. Moreover, the surface quality of A-plane sapphire after FIB milling is better than that of C-plane and M-plane sapphires. The theoretical calculation results show that the removal energy of aluminum ions and oxygen ions per square nanometer on the outermost surface of A-plane sapphire is the smallest. This also implies that material is more easily removed from the surface of A-plane sapphire than the surface of C-plane and M-plane sapphires by FIB milling. In addition, it is also found that higher MRR leads to lower Sa and better surface quality of sapphire for FIB etching.

## 1. Introduction

Sapphire (α-Al_2_O_3_) is a transparent, hard and brittle material that withstands high temperature, high pressure, and chemical corrosion [1,2]. It is well established that sapphire is an anisotropic material, and the three most commonly used crystal orientations are A ($11\bar{2}0$), C (0001), and M ($1\bar{1}00$). The arrangement of atoms on A-, C-, and M-oriented sapphire is different, which results in anisotropic optical properties [3,4], mechanical characteristics [5,6], thermal and chemical properties. Therefore, sapphire with the three different crystal orientations can be applied in different fields. For example, A-plane sapphire has been widely used in optical windows, microelectronics and high-temperature superconductors [7]. C-plane sapphire is often used as substrates of gallium nitride (GaN)-based light emitting diodes [8]. M-plane sapphire can be used to conduct the epitaxial growth of semipolar or nonpolar GaN films [9]. Previous studies have revealed that the material removal rate (MRR) and surface quality of single-crystal sapphire depend on the crystal orientation [10,11,12,13,14]. Therefore, the processing of anisotropy sapphire needs to consider the influence of crystal orientation on processing efficiency and quality. High precision micro/nano-structures on transparent substrates are highly desired for a wide range of applications for electrofluidic devices and optical devices fabrication [15]. The micro/nano-structures on sapphire substrate have many functions. For example, nanopatterned sapphire substrate plays an important role in enhancing the light output of GaN-based light emitting diodes [16]. Micro/nano-structures on sapphire can also be used to create a superhydrophobic surface for water repellence and self-cleaning applications [17]. There are several ways to manufacture micro/nano-structures on sapphire, such as machining, chemical etching, laser processing, and ion beam etching. However, micro/nano-structures on sapphire fabricated by machining has several defects, such as cracks, delamination, and surface damage [18]. Preparation of micro/nano-structures on sapphire by chemical corrosion has low efficiency, is environment unfriendly and harmful to users. Furthermore, the shape of micro/nano-structures on sapphire is also difficult to control precisely due to the anisotropic corrosion of the sapphire material. Laser processing can efficiently produce micro/nano-structures on sapphire. Nevertheless, laser-processed sapphire inevitably suffers from thermal damage, thermal cracking, and debris [13]. Focused ion beam (FIB) milling has proved to be a simple and efficient method for the fabrication of micro/nano-structures in the research field [19,20]. Since FIB etching removes material one atom at a time, the micro/nano-structures created by FIB milling have no surface damage, debris, or cracks. The major disadvantage of the FIB technique is throughput due to its extremely low etching rate. As a result, FIB milling is not suitable for industry. However, FIB technology still has many unique advantages, such as high resolution and flexibility, maskless processing, and rapid prototyping. Thus, FIB technology has become one of the key approaches in precision micro/nano-fabrication for various applications, including nano-optics, surface engineering, MEMS, bio-sensing, and nanotechnology [21,22].

In this study, rectangular pits were etched on A-, C-, and M-plane sapphires using a commercial FIB machine. The geometrical dimensions, as well as the etched volumes of the rectangular pits, were determined. Further, the MRRs of sapphire with A-, C-, and M-orientations were derived. To better understand the differences in MRR among the A-, C-, and M-orientations, the energy required to remove aluminum ions (Al^3+^) and oxygen ions (O^2−^) from the lattice sites at the outermost surface of sapphires with different orientations was theoretically calculated. In addition, the surface roughness of the bottom of the pits on sapphire substrates was characterized. The surface quality of sapphire with different crystal orientations after FIB etching was also analyzed.

## 2. Materials and Methods

Single-crystal sapphire specimens with A-, C-, and M-orientations were purchased from Helios New Materials Co., Ltd. (Jiangyin, China). The geometrical dimensions of these sapphire specimens were 10 mm in length, 10 mm in width and 0.43 mm in thickness. The surfaces of these sapphire specimens were polished, and their surface roughness (Sa) was about 0.4 nm. The A-plane ($11\bar{2}0$), C-plane (0001), and M-plane ($1\bar{1}00$) sapphires were oriented according to their axes, see Figure 1a–c.

A Seiko SMI3050 dual FIB system (SEIKO, Chiba, Japan) was used to etch rectangular pits on the polished surface of the A-, C-, and M-plane sapphires (Figure 1a–c). To compare the etched volumes of the rectangular pits on the A-, C-, and M-plane sapphires, the milling parameters for the sapphire specimens with three different crystal orientations were set to be identical. The FIB milling parameters were as follows: The ion beam accelerating voltage was 30 kV; the ion beam current was 800 pA; the pixel dwell time was 100 μs; the number of passes was 952; the total dosage of the ions was 18.78 nC/cm^2^; the etching area of each rectangular pit was set to 15 μm × 5 μm; and the processing time was set to 21 min for an area of 75 μm^2^. The arrows inside the pits in Figure 1a–c represent the milling direction of FIB. The morphologies of the rectangular pits on the A-, C-, and M-plane sapphires were characterized using field emission scanning electron microscopy (Supra 55, ZEISS, Jena, Germany). The etched geometrical dimensions and volumes, together with the surface roughness of the bottom of etched pits on A-, C-, and M-plane sapphires, were characterized using a three-dimensional (3D) optical surface profilometer (Newview 5032, ZYGO, Middlefield, CT, USA). The removal rate of the etched sapphire was determined by the removal volume per minute. The surface quality of the FIB-processed sapphire was evaluated for surface roughness.

## 3. Results and Discussion

### 3.1. Morphology and Dimension of the FIB-Etched Pits

SEM images of the rectangular pits on the A-, C-, and M-planes of the sapphire, with the samples tilted by 0°, are shown in Figure 2a–c, respectively. SEM images in Figure 2d–f were taken with the sapphire samples tilted by 30°. As can be seen, the FIB-etched pits have smooth and vertical sidewalls, and the bottom of the FIB-etched pit is also very smooth. Figure 3a,b depicts the cross-sectional depth profiles of the pits along the horizontal dashed lines in Figure 2a–c and the vertical dashed lines in Figure 2d–f. A 3D optical surface profilometer was used to measure the geometrical dimensions of the etched pits. The etched depth of the pit was about 1285.9 nm for A-plane, 1302.9 nm for C-plane, and 1291.6 nm for M-plane. The etched length of the pit was about 15.1 μm for A-pane, 14.7 μm for C-plane, and 14.8 μm for M-plane. The etched width of the pit was about 5.0 μm for A-plane, 4.5 μm for C-plane, and 4.7 μm for M-plane.

### 3.2. Material Removal Rates of A-, C-, and M-Plane Sapphires by FIB Milling

The material removal rate (MRR) is defined as the etched volume per minute. Hence, the MRR can be calculated by MRR = etched volume/processing time. Here the FIB processing time is 21 min. The etched volumes of the pits on the C- and M-planes of sapphire were also measured with a 3D optical surface profilometer and the results are depicted in Figure 4a. As can be seen in Figure 4a, the etched volume of the pit is around 97.17 μm^3^ for A-plane sapphire, 82.60 μm^3^ for C-plane sapphire, and 88.49 μm^3^ for M-plane sapphire, respectively. This indicates that the etched volume of the pit for the A-plane sapphire is larger than that for the C-plane and M-plane sapphires, while the etched volume of the pit for the C-plane sapphire is the smallest. According to the etched volumes of the pits on sapphire with different crystal orientations (Figure 4a), the MRRs of sapphire substrates including A-, C-, and M-planes can be derived, and the results are presented in Figure 4b. It can be seen that the average MRR is ~4.63 μm^3^/min for A-plane sapphire, ~4.21 μm^3^/min for M-plane sapphire, and ~3.93 μm^3^/min for C-plane sapphire. This indicates that the MRR of A-plane sapphire is slightly higher than that of C-plane and M-plane sapphires for FIB milling. The orientation-dependent MRR may be caused by the different bond energies of the bonds Al–O, Al–Al, and O–O for A-, C-, and M-plane sapphires. The differences in MRRs for different crystal planes of sapphire substrates are not so obvious in our experiment. We suspect that the small differences in MRRs are related to the removal depth of the FIB etching. As the etched depth decreases, the differences in MRRs for different crystal planes of the sapphire substrates gradually increase. In the future, we will study the influence of removal depth on the MRRs of single-crystal sapphires with different orientations. We will focus on the MRR of sapphire when the etching depth is on the nanometer scale.

To reveal the crystal orientation effects on the MRR of sapphire, it is worthwhile to review the crystal structure of sapphire. Figure 5a shows the crystal structure of a hexagonal unit cell of α-Al_2_O_3_ [23]. A primitive cell of the sapphire crystal has lattice parameters *a* = *b* = 4.758 Å and *c* = 12.991 Å. According to Table II in Reference [24], the total potential energy of Al^3+^ and O^2−^ ions at the lattice sites for a perfect crystal of α-Al_2_O_3_ is −90.4 and −41.1 eV, respectively. Since the zero of potential energy in this table is for an ion removed to infinity, the energy required to remove an Al^3+^ ion from a lattice site on the sapphire surface is 45.2 eV, while the energy to remove an O^2−^ ion from a lattice site on the sapphire surface is 20.55 eV (see Reference [24]). Figure 5b–d shows the expanded two-dimensional (2D) arrangement of Al^3+^ and O^2−^ ions at the lattice sites on the surface of A-, C-, and M-plane sapphires. As can be seen in Figure 5b, for A-plane sapphire, the length and width of the selected region are 2.472 and 2.599 nm, respectively, so the area of the selected region on A-plane sapphire surface is 6.42 nm^2^. The total numbers of Al^3+^ and O^2−^ ions within the selected region are 35 and 14, respectively. Thus, the total potential energy required to remove these ions in the selected region from the A-plane sapphire surface is 35 × 45.2 + 14 × 20.55 = 1869.70 eV. Thus, the energy required to remove Al^3+^ and O^2−^ ions per nm^2^ from A-plane sapphire surface is 291.23 eV/nm^2^. The energy required to remove Al^3+^ and O^2−^ ions per nm^2^ was calculated as 908.63 and 471.65 eV/nm^2^ for C- and M-planes of sapphire, respectively. Table 1 summarizes the removal energy of Al^3+^ and O^2−^ ions from square nanometers of the outermost surface of sapphires with different crystal planes. As can be seen in Table 1, the removal energy of Al^3+^ and O^2−^ ions per nm^2^ of the surface of A-, C-, and M-plane sapphires increases in the following order: *E*_A_ < *E*_M_ < *E*_C_. The larger the removal energy, the harder it is to remove material. This result also confirms that material is more easily removed from the surface of A-plane sapphire compared with the surface of C-plane and M-plane sapphires using FIB milling. However, it should be noted that our approach is based on a simplified 2D model. Only the Al^3+^ and O^2−^ ions on the outermost surface of sapphire are considered for the calculation of the removal energy per nm^2^.

### 3.3. Surface Roughness of the Pits on A-, C-, and M-Plane Sapphires

It can be seen intuitively in Figure 2a–f that the etched rectangular pits have outstanding surface quality. The Sa values of the bottom of the pits on A-, C-, and M-plane sapphires are shown in Figure 6. It was found that the surface roughness of the bottom of the pit was ~2.29 nm for C-plane sapphire, ~2.15 nm for M-plane sapphire, and ~1.01 nm for A-plane sapphire. This result illustrates that the surface quality of A-plane sapphire after FIB etching was better that that of C-plane and M-plane sapphires. The fact that A-plane sapphire has higher MRR and smaller Sa values compared with C-plane and M-plane sapphires for FIB milling is consistent with a previous report [14]. It was found from References [12,14] that when the material removal rate of sapphire is higher, the surface roughness is smaller. This phenomenon was also observed in our work.

## 4. Conclusions

In summary, we mainly studied FIB milling of single-crystal sapphire with A-, C-, and M-orientations. The experimental results show that the MRR of A-plane sapphire is slightly higher than that of the other two planes of sapphire. To clarify the crystal orientation effects on the MRR of sapphire, a simplified 2D model was used to calculate the energy required to remove Al^3+^ and O^2−^ ions per nm^2^ from the outermost surface of sapphire. The calculated removal energy of Al^3+^ and O^2−^ ions per nm^2^ is 291.23 eV/nm^2^ for A-plane, 908.63 eV/nm^2^ for C-plane, and 471.65 eV/nm^2^ for M-plane. This indicates that material is more easily removed from the surface of A-plane sapphire compared with the surface of C-plane and M-plane sapphires. The surface roughness of the bottom of the FIB-etched pit for A-oriented sapphire is 1.01 nm, which is smaller than that for C-oriented and M-oriented sapphires. This reveals that the surface quality of A-plane sapphire after FIB etching is better than that of C-plane and M-plane sapphires. It is also found that when the MRR of sapphire is higher, the surface roughness is smaller and the surface quality is better.

## Figures and Tables

**Figure 1 materials-13-02871-f001:**
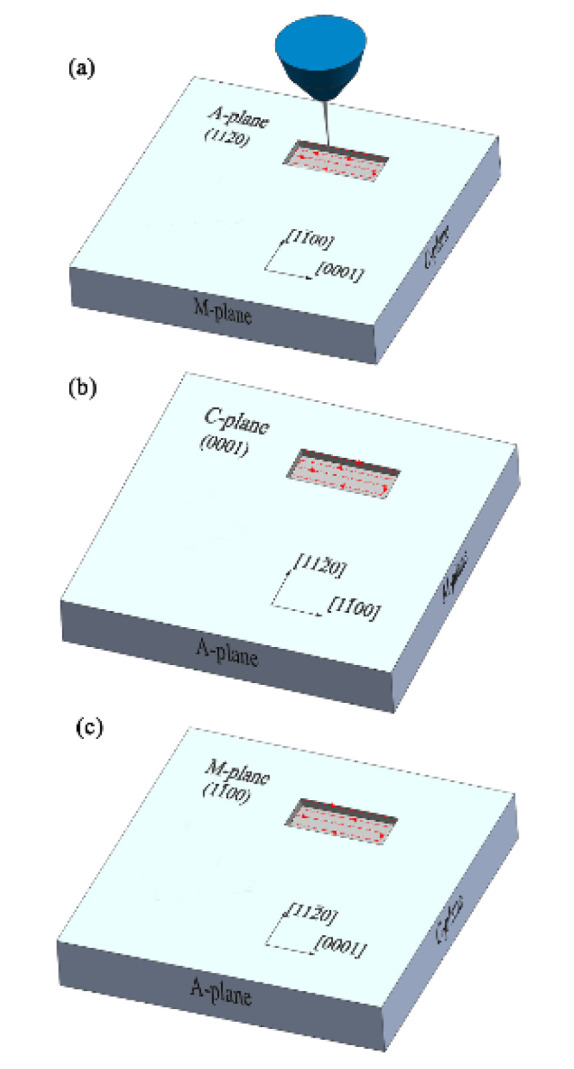
Schematic of rectangular pits on A-plane (**a**), C-plane (**b**), and M-plane (**c**) sapphires fabricated by focused ion beam (FIB) milling.

**Figure 2 materials-13-02871-f002:**
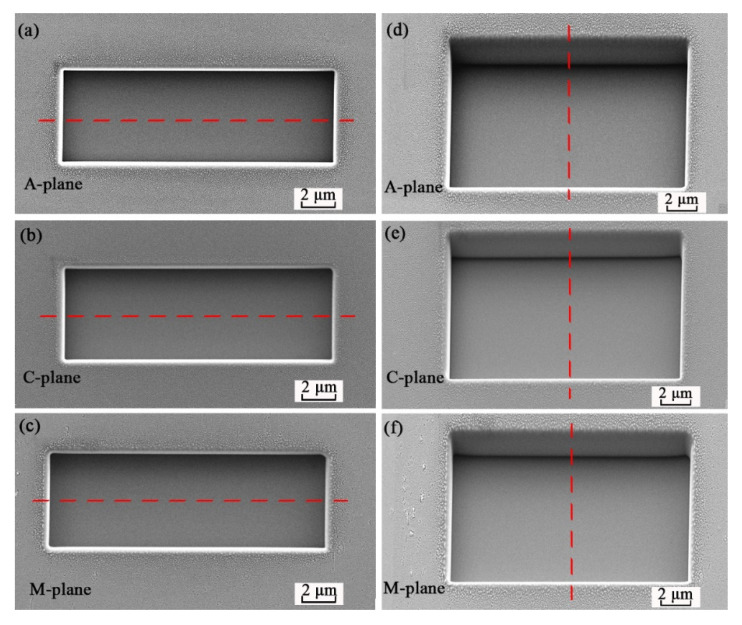
SEM images of the pits on the sapphire with different orientations taken at a sample tilt angle of 0° (**a**–**c**) and 30° (**d**–**f**), respectively.

**Figure 3 materials-13-02871-f003:**
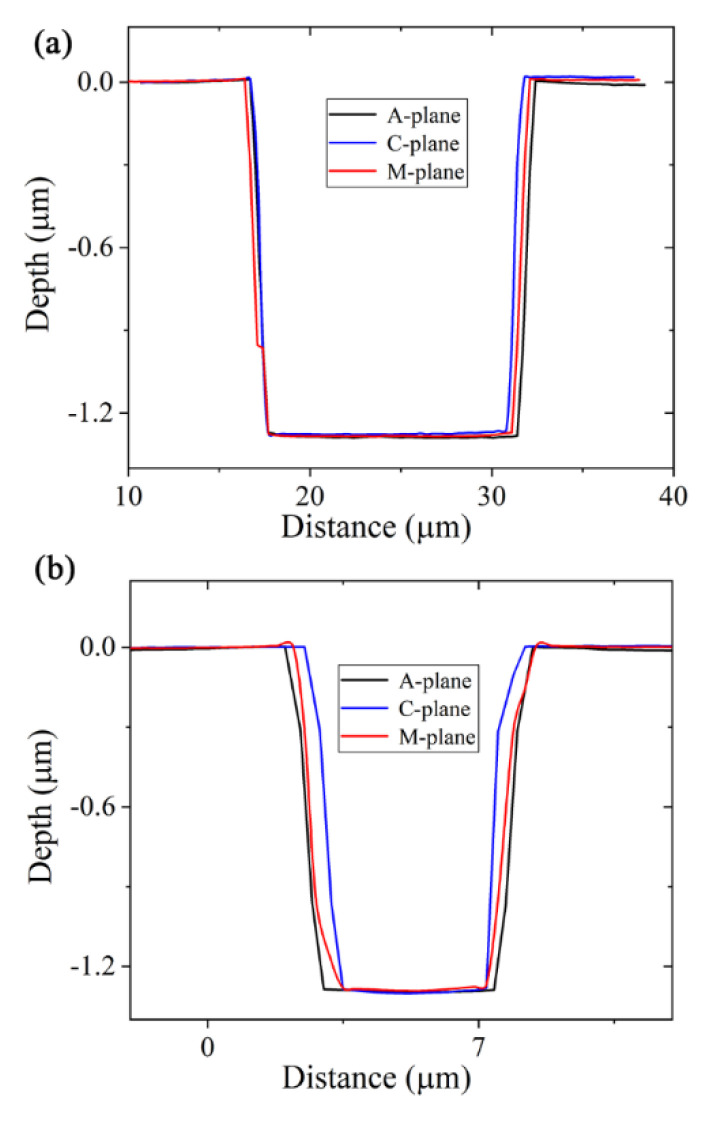
Cross-section depth profiles of etched pits along the horizontal direction (**a**) and vertical direction (**b**), respectively.

**Figure 4 materials-13-02871-f004:**
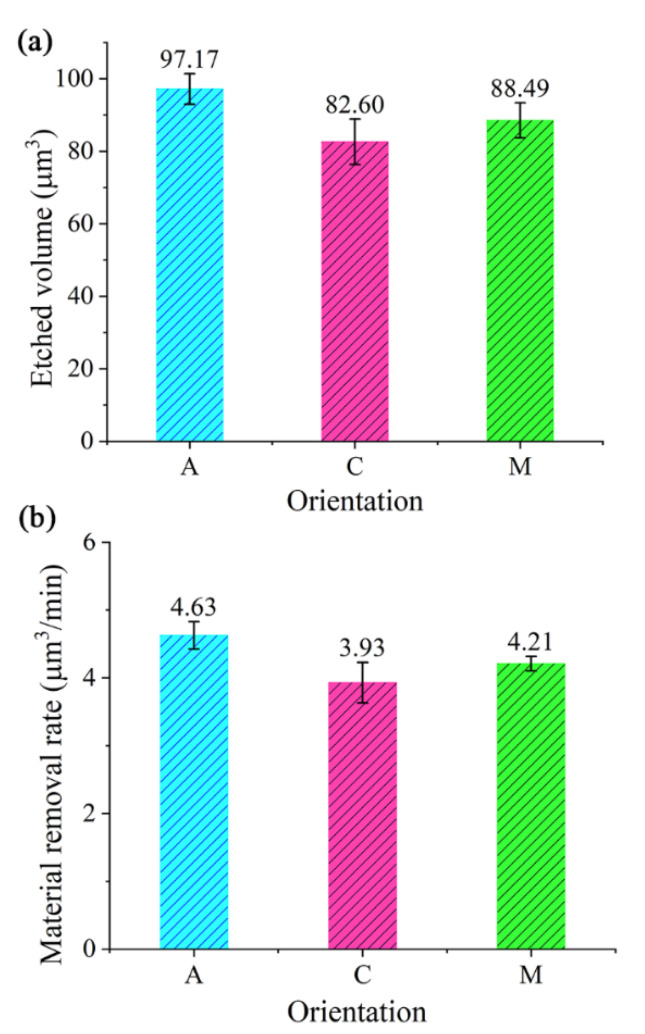
Etched volumes (**a**) and material removal rate (**b**) of the pits on A-, C-, and M-planes of sapphire.

**Figure 5 materials-13-02871-f005:**
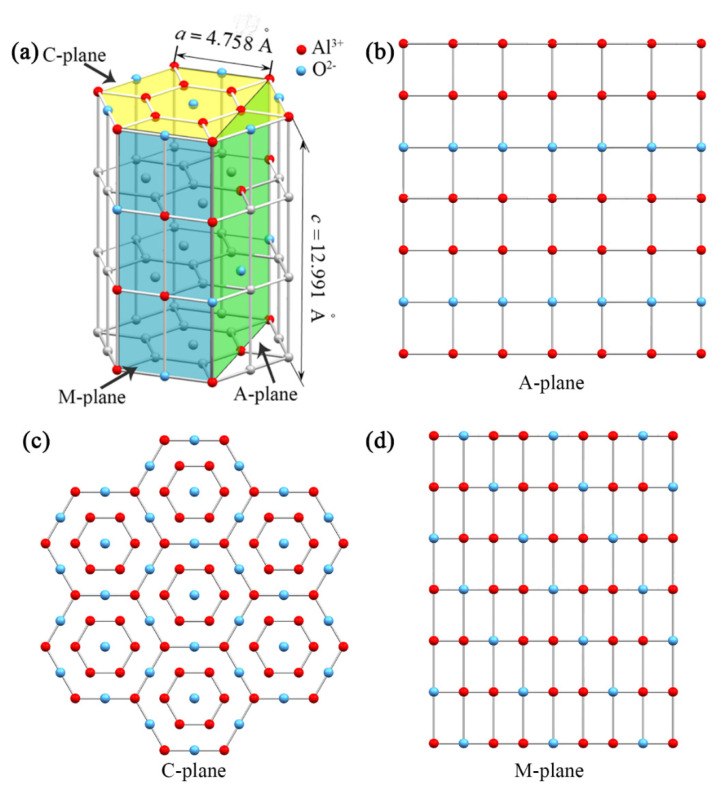
Crystal structure of sapphire (**a**) and expanded 2D arrangement of Al^3+^ and O^2−^ ions on the surface of A-plane (**b**), C-plane (**c**), and M-plane (**d**) sapphires.

**Figure 6 materials-13-02871-f006:**
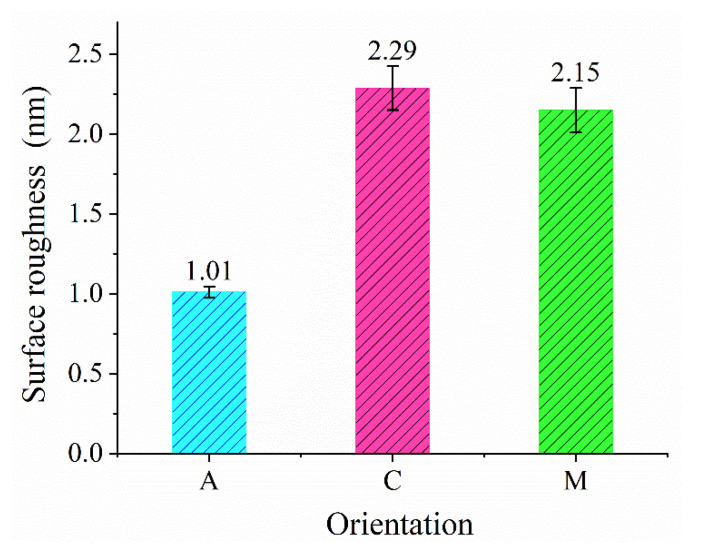
Surface roughness of the bottom of the FIB-etched pits on A-, C-, and M-plane sapphires.

**Table 1 materials-13-02871-t001:** Removal energy of Al^3+^ and O^2−^ ions per nm^2^ on the sapphire surface.

Orientation	Number of Al^3+^ Ions	Number of O^2−^ Ions	Area (nm^2^)	Total Potential Energy (eV)	Removal Energy Per nm^2^ for Surface Ions (eV/nm^2^)
A	35	14	6.42	1869.70	291.23
C	66	37	4.12	3743.55	908.63
M	42	21	4.94	2329.95	471.65

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
