# Peer review of "Focused Ion Beam Milling of Single-Crystal Sapphire with A-, C-, and M-Orientations"

_materials, 2020, doi:10.3390/ma13122871_

Round 1
Reviewer 1 Report
This work investigates the milling of different crystal orientations of sapphire by focused ion beam milling. The methodology is broadly sound and the conclusions are reasonable but both should be improved with some additional work.
- Line 27: The phrase “As we all known” should not be used. As a possible alternative consider: “It is well established that…”
- “Etching volume” should be replaced with “etched volume”
- Errors bars are needed for etched volume, material removal rates and surface roughness data. This would be useful considering the small differences between material removal rates of different crystal faces
- I have some concerns about the extent of the experimental data. As it stands, it seems the amount of experimental evidence is limited. Were results reproducible? It seems like these are singular observations. Was any attempt made to replicate the results or to provide some statistical weight through averaging of several experiments?
- Re-deposition of milled material can lead to non-linear etching rates for long FIB milling times, as can steep sidewalls for deep milled trenches. An evolution of etched volume with time would be a welcome addition. For example varying the milling time to be shorter than 21 minutes.
- The “theoretical” calculations are too simplistic, for example the energy required to remove the first atom and the last atom from the 2D region will not be equivalent. At best the current estimates in Table 1 (e.g. Removal energy per square nanometer) can only be considered to be upper bounds. The text should be modified to reflect the limitations of this approach.
Author Response
Response to Reviewer 1 Comments
Point 1: Line 27: The phrase “As we all known” should not be used. As a possible alternative consider: “It is well established that…”
Response 1: We are very sorry for our negligence of some errors in grammar. The phrase “As we all known” is corrected to “It is well established that” (Line 3 on Page 3 in the revised manuscript).
Point 2: “Etching volume” should be replaced with “etched volume”
Response 2: Thanks for the valuable comments. The phrase “Etching volume” is corrected to “etched volume” in the revised manuscript. We have checked the grammar of the entire manuscript, and some corrections are also highlighted in red.
Point 3: Errors bars are needed for etched volume, material removal rates and surface roughness data. This would be useful considering the small differences between material removal rates of different crystal faces
Response 3: Thanks for your valuable suggestions. We added the errors bars for etched volume, material removal rates and surface roughness data in Figures 4 and 6 (Page 11, Figure 4; Page 14, Figure 6).
Point 4: I have some concerns about the extent of the experimental data. As it stands, it seems the amount of experimental evidence is limited. Were results reproducible? It seems like these are singular observations. Was any attempt made to replicate the results or to provide some statistical weight through averaging of several experiments?
Response 4: Thanks for your comments. Actually, we have done this experiment three times, and three pits were etched on each crystal face. The results were average values of three pits on every specimen. Errors bars for etched volume, material removal rates and surface roughness data are added in Figures 4 and 6.
Point 5: Re-deposition of milled material can lead to non-linear etching rates for long FIB milling times, as can steep sidewalls for deep milled trenches. An evolution of etched volume with time would be a welcome addition. For example varying the milling time to be shorter than 21 minutes.
Response 5: Thanks for the valuable comments. The etched depth of the pits is shallow in our experiment, thus the effect of re-deposition of milled material on etching rates can be ignored. We intend to add an evolution of etched volume with milling time. Unfortunately, due to the outbreak of COVID-19, the laboratory is closed and we are not allowed to go back to the laboratory to do FIB milling.
Point 6: The “theoretical” calculations are too simplistic, for example the energy required to remove the first atom and the last atom from the 2D region will not be equivalent. At best the current estimates in Table 1 (e.g. Removal energy per square nanometer) can only be considered to be upper bounds. The text should be modified to reflect the limitations of this approach.
Response 6: Thanks for your comments. The theoretical calculation for removal energy per square nanometer in Table 1 is based on a simplified 2D model, which only considers the aluminum ions and oxygen ions on the outermost surface of sapphire. The limitations of this approach has been pointed out in the revised manuscript. (Lines 19-21, Page 12)

Reviewer 2 Report
Review of mdpi materials 828179 v1
The manuscript entitled “Focused ion beam milling of single-crystal sapphire
with A-, C-, and M-orientations” describes the milling properties of three different crystallographic planes in single-crystal alpha-alumina. The manuscript is missing some vital information and is not ready for publication at this stage. After a major revision that addresses the following points, it might be reconsidered.
- The manuscript requires some significant English language improvements. It is written in certain places in a very colloquial way. For instance, “As we all known [sic], …” is not an appropriate statement for a scientific manuscript. First of all, it is grammatically wrong and second who are “we”? is it the reader, is it the author? This is just an example; please improve and correct these types of language issues.
- The quality of the images needs significant improvement. None of the figures are of publication quality. They have very low definition and exhibit compression artifacts.
- In figures 2a and 2b, we see the SEM image of the etch pit on A-plane sapphire, but the reader would be interested to see the SEM images of the etch pits on other surfaces as well.
- The graphs and manuscript give significant figures sometimes in the first decimal and sometimes to the second decimal without any statistical analysis. Without knowing what the experimental errors are, it is very difficult to take these high precision measurements as the complete truth.
- Therefore, I suggest the authors repeat the measurements and provide at least pattern to pattern variations (if not sample to sample). I would like to see the errors on those measurements, and please make sure to note what the significant figures are.
- I have a more fundamental question; the z-resolution of the optical surface profiler is very high, while the lateral resolution is limited by the wavelength used. It certainly lacks the lateral resolution of an AFM. Therefore, the nm-scale surface roughness values must come from the height differences over relatively large areas. I think this has to be pointed out as a weakness of the optical profiler method. It would be interesting to see the AFM measurements of the surface roughness and how it compares to the optical profiler results.
- I am missing some essential information in the FIB milling parameters, such as the pixel dwell time, the number of passes, and the total dosage of the ions. Without this info, it is challenging to know what is going on with the milling response of the material.
Therefore, I suggest a resubmission after a major revision of the work.
Author Response
Response to Reviewer 2 Comments
Point 1: The manuscript requires some significant English language improvements. It is written in certain places in a very colloquial way. For instance, “As we all known [sic], …” is not an appropriate statement for a scientific manuscript. First of all, it is grammatically wrong and second who are “we”? is it the reader, is it the author? This is just an example; please improve and correct these types of language issues.
Response 1: Thanks for your comments. Manuscript has been thoroughly checked for grammatical and format errors. The grammatical and typographical errors, such as spelling, case of letters and tense have been corrected and marked in red. The phrase “As we all known” has been revised to “It is well established that…”. (Line 3 on Page 3 in the revised manuscript).
Point 2: The quality of the images needs significant improvement. None of the figures are of publication quality. They have very low definition and exhibit compression artifacts.
Response 2: Thanks for the valuable comments. We have improved the quality of the images and ensured that these images can meet the requirements of publishing quality.
Point 3: In figures 2a and 2b, we see the SEM image of the etch pit on A-plane sapphire, but the reader would be interested to see the SEM images of the etch pits on other surfaces as well.
Response 3: Thanks for your suggestions. The SEM images of the etch pits on C-plane and M-plane sapphires have been added in Fig. 2.
Point 4: The graphs and manuscript give significant figures sometimes in the first decimal and sometimes to the second decimal without any statistical analysis. Without knowing what the experimental errors are, it is very difficult to take these high precision measurements as the complete truth. Therefore, I suggest the authors repeat the measurements and provide at least pattern to pattern variations (if not sample to sample). I would like to see the errors on those measurements, and please make sure to note what the significant figures are.
Response 4: Thanks for your comments. The etched volume, surface removal energy and surface roughness are obtained by indirect calculation, and the calculation results uniformly keep two decimal places. The 3D optical surface profiler used in our experiment has a lateral resolution of 200 nm for the x and y axes, and a z-resolution of 0.1 nm. The measurement accuracy of etch length and etch width are determined by the lateral resolution of optical surface profiler. While the measurement accuracy of etch depth is determined by z-resolution of optical surface profiler. We repeat the measurements for many times, and error bars have been added in Figures 4 and 6.
Point 5: I have a more fundamental question; the z-resolution of the optical surface profiler is very high, while the lateral resolution is limited by the wavelength used. It certainly lacks the lateral resolution of an AFM. Therefore, the nm-scale surface roughness values must come from the height differences over relatively large areas. I think this has to be pointed out as a weakness of the optical profiler method. It would be interesting to see the AFM measurements of the surface roughness and how it compares to the optical profiler results.
Response 5: Thanks for the valuable comments. The lateral resolution of the optical surface profiler is 200 nm. Therefore, the lateral measurement accuracy of optical surface profiler is indeed lower than that of AFM. Optical surface profiler is often used to measure the surface roughness for relatively large areas. This is indeed a weakness of the optical profiler method. But we think optical surface profiler can be used to measure the surface roughness of the bottom of the etched pits on sapphire, and the measurement results are also reliable. It will be more convincing if the AFM measurements of the surface roughness of the etched pits could be provided. Unfortunately, due to the outbreak of COVID-19, our laboratory is closed and we are not allowed to go back to the laboratory to do AFM characterization.
Point 6: I am missing some essential information in the FIB milling parameters, such as the pixel dwell time, the number of passes, and the total dosage of the ions. Without this info, it is challenging to know what is going on with the milling response of the material.
Response 6: Thanks for your comments. Detailed FIB milling parameters, such as the pixel dwell time, the number of passes, and the total dosage of the ions are included in Section 2 (Materials and Methods, Lines 4-5 of Page 7 in the revised manuscript).

Reviewer 3 Report
In this paper the authors etch sapphire with different orientatons by 30 kV Ga+ ions and reveal the differences in etch rate and quality.
Major issues:
- The paper begins with the statement that sapphire is anisotropic. I should note that any crystal is anisotropic, e.g. Si with diamond structure. Si also has different etch rates for different orientations for the same reasons. Therefore, the obtained results are rather trivial. These results could be interesting only for those who etch sapphire and not for a broad material science community.
- In lines 39-40 the authors claim "It is of great industrial value to fabricate micro/nano-structures on sapphire substrate." Then the authors explain disadvantages of different approaches to sapphire etching (mechanical and wet chemistry, laser), however they completely miss the main disadvantage of their own approach (FIB), i.e. extremely low etching rates. For this reason the discussed method is not suitable for industry. Therefore the motivation for research is not clear.
Minor remarks:
- Lines 36-37. The sentence "Therefore, processing isotropic sapphire needs to consider the influence of crystal orientation." is not clear
- Materials and methods. Probably a commercial FIB maschine was used. Which one? Please indicate
I remind the authors that "Materials" journal publishes the papers "which advance the in-depth understanding of the relationship between the structure, the properties or the function." The presented results are trivial and very narrow. Moreover, the authors observe a rather small orientation effect on etch rate (much smaller than their own energy expectation). Therefore, after minor improvements I suggest to resubmit this paper to a more specialized journal, e.g. MDPI Applies Science or Electronics or Electronic Materials.
Author Response
Response to Reviewer 3 Comments
Point 1: The paper begins with the statement that sapphire is anisotropic. I should note that any crystal is anisotropic, e.g. Si with diamond structure. Si also has different etch rates for different orientations for the same reasons. Therefore, the obtained results are rather trivial. These results could be interesting only for those who etch sapphire and not for a broad material science community.
Response 1: Thanks for your comments. Single crystal sapphire has a hexagonal-rhombohedral crystal structure. This paper has reference value for the processing of crystals with similar structure.
Point 2: In lines 39-40 the authors claim "It is of great industrial value to fabricate micro/nano-structures on sapphire substrate." Then the authors explain disadvantages of different approaches to sapphire etching (mechanical and wet chemistry, laser), however they completely miss the main disadvantage of their own approach (FIB), i.e. extremely low etching rates. For this reason the discussed method is not suitable for industry. Therefore the motivation for research is not clear.
Response 2: Thanks for your comments. The major disadvantage of FIB technique is throughput due to its extremely low etching rate. However, it is very convenient to fabricate the desired micro/nano-structures on sample material locally by direct FIB milling without the need of a resist layer and a mask [Ref. 20]. In the semiconductor industry, FIB is widely used for integrated circuit repair [Refs. 21 and 22]. It is very valuable to use FIB milling to repair defects for new chip design because it can save time and iteration from a new design to a working part. (Lines 13-19, Page 4)
Minor remarks:
Point 1: Lines 36-37. The sentence "Therefore, processing isotropic sapphire needs to consider the influence of crystal orientation." is not clear
Response 1: Thanks for the comments. The sentence "Therefore, processing isotropic sapphire needs to consider the influence of crystal orientation." has been revised to “Therefore, the processing of anisotropy sapphire needs to consider the influence of crystal orientation on processing efficiency and quality”. (Lines 15-16 on Page 3)
Point 2: Materials and methods. Probably a commercial FIB machine was used. Which one? Please indicate
Response 2: Thanks for the comments. In our experiment, a Seiko SMI3050 FIB machine was used to etch pits on sapphire. This has been indicated in Line 3 on Page 6 of the revised manuscript.

Round 2
Reviewer 1 Report
I still believe this work could be improved but given the current climate it is acceptable.
Author Response
Thanks for your comments.
Reviewer 2 Report
I suggest acception in the current form.
In the 2nd version I got, I still see artifacts in images. But this might be due to the pdf conversion during the submission process. Just wanted to hint at this issue again.
Author Response
Thanks for your suggestions.
Reviewer 3 Report
Dear editors,
Although the text has been improved, my major queries were not answered. The main concern was that the topic is too narrow (and opinion of the other referee was the same). In the rebuttal the authors try to explain that FIB technique is widely used in semiconductor industry (lines 58-59 in the revised manuscript):
"In the semiconductor industry, FIB is widely used for integrated circuit repair [21,22]."
This statement is quite confusing, because integrated circuits are not used to be repaired (at least industrially). The given references also do not support the statement about wide use of the FIB.
FIB is the tool for research, not for industry.
I completely agree with the other referee that the differences in material removal rates for different substrates are not much different. So there is no interesting result.
Moreover, I do not understand why these rates should be different, because the same volumes have the same amounts of the chemical bonds. To break these bonds the same energy is needed.
Therefore, the paper does not contain valuable information for the scientific community, and I still insist that it should be published in a more specialized journal, not in MDPI Materials.
Author Response
Response to Reviewer 3 Comments
Point 1: In the rebuttal the authors try to explain that FIB technique is widely used in semiconductor industry (lines 58-59 in the revised manuscript): "In the semiconductor industry, FIB is widely used for integrated circuit repair [21,22]." This statement is quite confusing, because integrated circuits are not used to be repaired (at least industrially). The given references also do not support the statement about wide use of the FIB. FIB is the tool for research, not for industry.
Response 1: Thanks for your comments. We completely agree with you that FIB technology is indeed used in the research field and is not suitable for industry due to its extremely low etching rate. However, FIB technology still has many unique advantages such as high resolution and flexibility, maskless processing, and rapid prototyping. FIB technology has become one of the key approaches in the precision micro/nanofabrication for various applications, including, nano-optics, surface engineering, MEMS, bio-sensing, and nanotechnology. We have revised the description in the revised manuscript (Line 11 and Lines 13-19 on Page 4) and changed references 21 and 22 (Page 19).
Point 2: I completely agree with the other referee that the differences in material removal rates for different substrates are not much different. So there is no interesting result.
Response 2: Thanks for your comments. We admit that the differences in material removal rates for different crystal planes of sapphire substrates are not so obvious in our experiment. However, the theoretical results show that the differences in the removal energies of surface ions for different crystal planes of sapphire substrates is still very obvious. We suspect that the small differences in material removal rates for different crystal planes of sapphire substrates is related to the removal depth of FIB etching. As the etched depth decreases, the difference in material removal rates for crystal planes of sapphire substrates will gradually increase. In the further, we will study the influence of removal depth on the material removal rates of single crystal sapphire with different orientations. We will focus on the material removal rate when the etching depth is on the nanometer scale. (Lines 10-16, Page 10)
Point 3: Moreover, I do not understand why these rates should be different, because the same volumes have the same amounts of the chemical bonds. To break these bonds the same energy is needed.
Response 3: Thanks for your comments. As can be seen from Fig.5 in this work and Fig. 11 in our previous work [13], the length of bonds Al–O, Al–Al, and O–O is different for A-, C- and M-planes of sapphire. Therefore, the bond energy of Al–O, Al–Al, and O–O is also different for A-, C- and M-planes of sapphire. Although the same volumes contain the same amounts of the chemical bonds, the bond energies of these chemical bonds are different, thus, the energy required to break these bonds is different, which results in different material removal rates (Lines 8-10, Page 10). Just like the meat cutting, the difficulty of cutting meat from different directions is different.
Round 3
Reviewer 3 Report
The authors placed their research into the context more accurately. My querry was addressed.